# RepPoints v2: Verification Meets Regression for Object Detection

**Yihong Chen**[*][14]**, Zheng Zhang**[2]**, Yue Cao**[2]**, Liwei Wang**[13]**, Stephen Lin**[2]**, Han Hu**[2]

[1]Center of Data Science, Peking University
[2]Microsoft Research Asia
[3]Key Laboratory of Machine Perception, MOE, School of EECS, Peking University
[4]Zhejiang Lab
{v-yich,zhez,yuecao,stevelin,hanhu}@microsoft.com
wanglw@cis.pku.edu.cn

## Abstract

Verification and regression are two general methodologies for prediction in neural networks. Each has its own strengths: verification can be easier to infer accurately, and regression is more efficient and applicable to continuous target variables. Hence, it is often beneficial to carefully combine them to take advantage of their benefits. In this paper, we take this philosophy to improve state-of-the-art object detection, specifically by RepPoints. Though RepPoints provides high performance, we find that its heavy reliance on regression for object localization leaves room for improvement. We introduce verification tasks into the localization prediction of RepPoints, producing RepPoints v2, which provides consistent improvements of about 2.0 mAP over the original RepPoints on the COCO object detection benchmark using different backbones and training methods. RepPoints v2 also achieves 52.1 mAP on COCO `test-dev` by a single model. Moreover, we show that the proposed approach can more generally elevate other object detection frameworks as well as applications such as instance segmentation. The code is available at https://github.com/Scalsol/RepPointsV2.

## 1 Introduction

Two common methodologies for neural network prediction are verification and regression. While either can drive network features to fit the final task targets, they each have different strengths. For the object localization problem, verification can be easier to infer because each feature is spatially aligned with the target to be verified. On the other hand, regression is often more efficient and it can also predict continuous target variables that enable subtle localization refinement.

To take advantage of all these benefits, earlier object localization methods [9, 20, 19] combined verification and regression by first performing coarse localization through verifying several anchor box hypotheses, and then refining the localization by regressing box offsets. This combination approach was shown to be effective and led to state-of-the-art performance at the time. However, recent methods based purely on regression, which directly regress the object extent from each feature map position [32, 29, 34], could perform competitively or even better, when comparing a representative regression method, RepPoints, to RetinaNet [19].

In this work, we examine whether pure regression based methods can be enhanced by the inclusion of verification methodology. We observe that verification has proven to be advantageous when used in certain ways. In CornerNet [16], feature map points are verified as a bounding box corner or

---

[*]Most work was done when Yihong Chen was an intern at Microsoft Research Asia.

not, in contrast to verifying anchor boxes for coarse hypothesis localization in RetinaNet [19]. This use of verification leads to significantly better localization performance as shown in Table 1. The difference may be attributed to corner points representing the exact spatial extent of a ground-truth object box, while an anchor box gives only a coarse hypothesis. In addition, each feature in corner point verification is well aligned to the corresponding point, while in anchor verification, the center feature used for verification lies away from the boundary area.

To elevate the performance of regression-based methods, specifically RepPoints [32], we thus seek to incorporate effective and compatible forms of verification. However, the different granularity of object representations processed by the two methods, i.e., whole objects in RepPoints and object parts (corners) in corner verification, presents an obstacle. To address this issue, we propose to model verification tasks by *auxiliary side-branches* that are added to the major regression branch at only the feature level and result level, without affecting intermediate representations. Through these auxiliary side-branches, verification can be fused with regression to provide the following benefits: better features by multi-task learning, feature enhancement through inclusion of verification cues, and joint inference by both methodologies. The fusion is simple, intuitive, general enough to utilize any kind of verification cue, and does not disrupt the flow of the RepPoints algorithm.

Through different techniques for harnessing verification, the localization and classification ability of RepPoints is substantially improved. The resulting detector, called RepPoints v2, shows consistent improvements of about 2.0 mAP over the original RepPoints on the COCO benchmark with different backbones. It also achieves 52.1 mAP on the COCO object detection `test-dev` set with a single ResNeXt-101-DCN model.

The proposed approach of choosing proper verification tasks and introducing them into a regression framework as *auxiliary* branches is flexible and general. It can be applied to object detection frameworks other than RepPoints, such as FCOS [29]. The additional corner and within-box verification tasks are shown to improve a vanilla FCOS detector by 1.3 mAP on COCO `test-dev` using a ResNet-50 model. This approach can be also applied beyond object detection, such as to instance segmentation by Dense RepPoints [33], where additional contour and mask verification tasks improve performance by 1.3 mAP using a ResNet-50 model on the COCO instance segmentation `test-dev` set, reaching 38.9 mask mAP.

Table 1: Analysis of the performance on COCO `val` set among different methods. "RepPoints*" indicates our improved re-implementation of RepPoints.

| Method | methodology | backbone | AP | $AP_{50}$ | $AP_{60}$ | $AP_{70}$ | $AP_{80}$ | $AP_{90}$ |
|---|---|---|---|---|---|---|---|---|
| RetinaNet [19] | ver.+reg. | ResNeXt-101 | 40.0 | 60.9 | 56.4 | 48.7 | 35.8 | 14.6 |
| CornerNet [16] | verification | HG-104 | 40.6 | 56.1 | 52.0 | 46.8 | 38.8 | 23.4 |
| RepPoints* [32] | regression | ResNet-50 | 39.1 | 58.8 | 54.8 | 48.0 | 35.5 | 14.4 |
| RepPoints v2 | ver.+reg. | ResNet-50 | 41.0 | 59.9 | 55.9 | 49.1 | 37.2 | 18.5 |

## 2 Related Works

**Verification based object detection** Early deep learning based object detection approaches [28, 26] adopt a multi-scale sliding window mechanism to verify whether each window is an object or not. Corner/extreme point based verification is also proposed [30, 16, 37, 6, 5] where the verification of a 4-d hypothesis is factorized into sub-problems of verifying 2-d corners, such that the hypothesis space is more completely covered. A sub-pixel offset branch is typically included in these methods to predict continuous corner coordinates through regression. However, since this mainly deals with quantization error due to the lower resolution of the feature map compared to the input image, we treat these methods as purely verification based in our taxonomy.

**Regression based object detection** Achieving object detection by pure regression dates back to YOLO [22] and DenseBox [13], where four box borders are regressed at each feature map position. Though attractive for their simplicity, their accuracy is often limited due to the large displacements of regression targets, the issue of multiple objects located within a feature map bin, and extremely imbalanced positive and negative samples. Recently, after alleviating these issues by a feature pyramid network (FPN) [18] structure along with a focal loss [19], regression-based object detection has regained attention [29, 15, 36, 32], with performance on par or even better than other verification

or hybrid methods. Our work advances in this direction, by leveraging verification methodology into regression based detectors without disrupting its flow and largely maintaining the convenience of the original detectors. We mainly base our study on the RepPoints detector, but the method can be generally applied to other regression based detectors.

**Hybrid approaches** Most detectors are hybrid, for example, those built on anchors or proposals [8, 7, 23, 19, 1, 21]. The verification and regression methodologies are employed in succession, where the anchors and proposals which provide coarse box localization are verified first, and then are refined by regression to produce the detection output. The regression target is usually at a relatively small displacement that can be easily inferred. Our work demonstrates a different hybrid approach, where the verification and regression steps are not run in serial but instead mostly in *parallel* to better combine their strengths. Moreover, this paper utilizes the more accurate corner verification tasks to complement regression based approaches.

**Multi-task learning** Several methods [11, 5, 4] adopt multi-task learning and observe moderate gain compared to the baseline. In [11], performance elevation in box mAP is observed when the mask annotation is utilized for training. Similarly, [4] utilize keypoint prediction as an auxiliary task and also observe performance gain in the main task. As our work aims to take advantage of all benefits from verification and regression, thus our proposed method has a high relevance to multi-task learning. However, unlike these methods, our contribution is far beyond multi-task learning: we propose a unified and general framework with 3 mechanisms to take advantage of verification tasks to help regression methods. Moreover, our multi-task learning does not require additional annotation.

## 3 Verification Meets Regression for Object Detection

### 3.1 A Brief Review of a Pure Regression Method: RepPoints

RepPoints [32] adopts pure regression to achieve object localization. Starting from a feature map position $\mathbf{p} = (x, y)$, it directly regresses a set of points $\mathcal{R}' = \{\mathbf{p}'_i = (x'_i, y'_i)\}_{i=1}^n$ to represent the spatial extent of an object using two successive steps:

$$\mathbf{p}_i = \mathbf{p} + \Delta\mathbf{p}_i = \mathbf{p} + \mathbf{g}_i(F_{\mathbf{p}}), \quad \mathbf{p}'_i = \mathbf{p}_i + \Delta\mathbf{p}'_i = \mathbf{p}_i + \mathbf{g}'_i(\text{concat}(\{F_{\mathbf{p}_i}\}_{i=1}^n)), \quad (1)$$

where $\mathcal{R} = \{\mathbf{p}_i = (x_i, y_i)\}_{i=1}^n$ is the intermediate point set representation; $F_{\mathbf{p}}$ denotes the feature vector at position $\mathbf{p}$; $\mathbf{g}_i$ and $\mathbf{g}'_i$ are 2-d regression functions implemented by a linear layer. The bounding boxes of an object are obtained by applying a conversion function $\mathcal{T}$ on the point sets $\mathcal{R}$ and $\mathcal{R}'$, where $\mathcal{T}$ is modeled as the *min-max*, *partial min-max* or *moment* function.

The direct regression in RepPoints [32] makes it a simple framework without anchoring. Though no anchor verification step is employed, it performs no worse than anchor-based detectors, i.e. RetinaNet [19], in localization accuracy as shown in Table 1. Nevertheless, we are motivated by the potential synergy between regression and verification to consider the following questions: What kind of verification tasks can benefit the regression-based RepPoints [32]? Can various verification tasks be conveniently fused into the RepPoints framework without impairing the original detector?

### 3.2 Verification Tasks

We first discuss a pair of verification tasks that may help regression-based localization methods.

#### 3.2.1 Corner Point Verification

Two corner points, e.g. the top-left corner and bottom-right corner, can determine the spatial extent of a bounding box, providing an alternative to the usual 4-d descriptor consisting of the box's center point and size. This has been used in several bottom-up object detection methods [16, 37, 30], which in general perform worse than other kinds of detectors in classification, but is significantly better in object localization, as seen in Table 1. In later sections, we show that this verification task can complement regression based methods to obtain more accurate object localization.

Corner point verification operates by associating a score to each point in the feature map, indicating its probability of being a corner point. An additional offset is predicted to produce continuous coordinates for corner points, which are initially quantized due to the lower resolution of the feature map compared to the input image, e.g. $8\times$ downsampling. Following the original implementation [16],

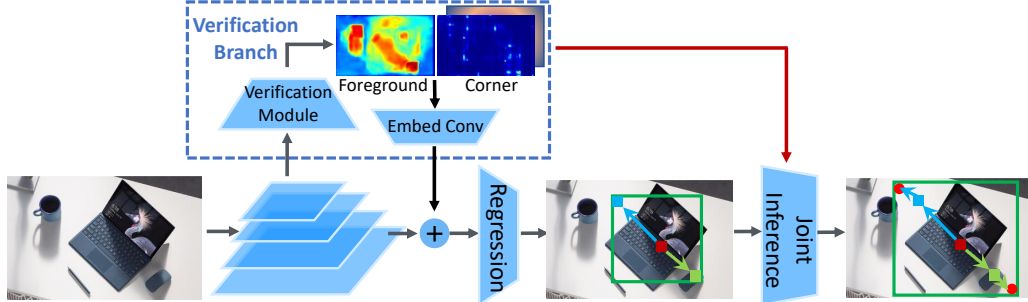

Figure 1: Overview of the general fusion method. The outputs of verification modules (corner and foreground) are incorporated with the input feature to elevate the performance of regression-based object localization, and then joint inference is further employed.

corner pooling is computed in the head, with a focal loss [19] to train the corner score prediction and a smooth L1 loss for the sub-pixel offset prediction. In label assignment, each feature map point is labeled positive if a ground truth corner point is located within its feature bin, and others are labeled negative. In computing the loss, the negative samples around each ground truth corner are assigned lower weights by an inverse Gaussian function with respect to its distance to the ground-truth corner point. A more detailed description is given in Appendix.

Different from CornerNet [16], which employs a special backbone architecture with an Hourglass structure and a single-level high resolution feature map ($4\times$ downsampled from the original image), most other recent object detectors adopt an FPN backbone with multi-level feature maps. We adapt the corner verification to utilize multi-level feature maps, e.g. the C3-C7 settings in RepPoints [32]. Specifically, all ground truth corner points are assigned to every feature map level, contrary to the usual practice in FPN-based object detection of assignment according to object size. We find that assignment in this manner performs slightly better although it disregards the scale normalization issue, probably due to more positive samples at each level in training. It also performs substantially better than training on a single feature map level of highest resolution, i.e. C3, and then copying/resizing the predicted score/offset map to other levels.

### 3.2.2 Within-box foreground verification

Another verification task with potential to benefit regression based object detectors is to verify whether a feature map point is located within an object box or not. This within-box foreground verification task provides localization information evenly inside an object box, in contrast to corner points which focus on the box extremes. It is thus not as precise as corner points in describing object bounds, but may benefit object detectors given a coarse localization criterion. The centerness prediction task in FCOS [29] shares some similarity with our within-box foreground verification task but they are designed for different roles. While FCOS's centerness aims for adjusting object scores and weighting the regression loss, the within-box verification aims to complement the main regression branch.

We also differentiate among different object categories by using a non-binary category-aware foreground heatmap. Concretely, for $C$ object categories, there is a $C$-channel output, with each channel indicating the probability of a feature point being in the corresponding object category. The same as for corner point verification, each ground truth object is assigned to every level of an FPN backbone.

**Normalized focal loss.**    In training, a vanilla focal loss lets larger objects contribute significantly more than smaller objects, resulting in poorly learnt foreground scores for small objects. To address this issue, a normalized focal loss is proposed, which normalizes every positive feature map point by the total number of positive points within the same object box in the feature map. For negative points, the normalized loss uses the number of positive points as the denominator. A more detailed description is given in Appendix.

### 3.3 A General Fusion Method

In this section, we incorporate these forms of verification to elevate the performance of regression-based methods. In general, regression-based methods detect objects in a top-down manner where all intermediate representations model the whole object. Since the two verification tasks process object parts, such as a corner or a foreground point, the different granularity of their object representations complicates fusion of the two methodologies.

To address this issue, we propose to model verification tasks by auxiliary side-branches that are fused with the major regression branch in a manner that does not affect its intermediate representations, as illustrated in Figure 1. Fusion occurs only at the feature level and result level. With these auxiliary side-branches, the detector can gain several benefits:

**Better features by multi-task learning**    The auxiliary verification tasks provide richer supervision in learning, yielding stronger features that increase detection accuracy, as shown in Table 4. Note that this multi-task learning is different from that of Mask R-CNN [10]. In Mask R-CNN [10], the bounding box object detection benefits from the object mask prediction task, but it requires extra annotation of the object mask. In contrast, our additional auxiliary tasks are automatically generated from only the object bounding box annotation, allowing them to be applied in scenarios where just bounding box annotations are available.

**Feature enhancement for better regression**    The verification output includes strong cues regarding corner locations and the foreground area, which should benefit the regression task. Since the prediction output of these verification tasks has the same resolution as the feature map used for regression at each FPN level, we directly fuse them by applying a *plus* operator on the original feature map and an embedded feature map produced from the verification output by one $1 \times 1$ conv layer. The embedding aims to project any verification output to the same dimension as the original feature map, and is shared across feature map levels. Note that for the verification output, a copy detached from back-propagation is fed into the embedding convolution layer to avoid affecting the learning of that verification task.

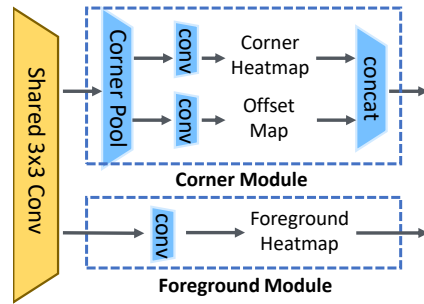

Figure 2: Illustration of the corner module and foreground module.

**Joint inference by both methodologies**    Feature-level fusion implicitly aids object localization. We also explicitly utilize the verification output from corner prediction together with regression-based localization in a joint inference approach that makes use of both of their strengths. Specifically, by corner verification, the sub-pixel corner localization in a small neighborhood is usually more accurate than that by the main regression branch, but is worse at judging whether it is a real corner point since it lacks a whole picture of the object. On the contrary, the main regression branch is better for the latter while worse in accurate sub-pixel localization. To combine their strengths, we refine a corner point $\mathbf{p}^t$ of the bounding box predicted from the main regression branch by

$$\text{refine}(\mathbf{p}^t) = \underset{\left\{\mathbf{q}^t \middle| \|\mathbf{q}^t - \mathbf{p}^t\| \leq r\right\}}{\arg\max} s(\mathbf{q}^t), \tag{2}$$

where $t$ indicates the corner type (top-left or bottom-right); $\mathbf{q}^t$ is a sub-pixel corner point produced by a corner prediction branch at a feature map position; $s(\mathbf{q}^t)$ is the verification score; $r$ is the neighborhood threshold, set to 1 by default. Unlike in training where we assign ground truth corners to multiple levels, C3-C7, of the feature pyramid, in joint inference only the C3 corner heatmap is utilized for searching. So $r = 1$ corresponds to a 8-pixel neighborhood. We find utilizing the heatmap of the highest resolution performs better than using the corresponding heatmap level. We hypothesis the reason is that higher resolution would produce more accurate corner prediction results, just in accordance of the observation in CornerNet [16]. So this design choice is more suitable for result-level fusion. Note that this result-level fusion is designed for the corner verification task only.

This fusion method is flexible and general, utilizing any kind of verification cue, as it avoids interaction with the intermediate representations in the main branch, and thus has few requirements on the types of verification target. It also does not disrupt the overall flow of the main branch and largely maintains the convenience of the original detector built on the main branch.

### 3.4 RepPoints v2: Fusing Verification into RepPoints

RepPoints is a pure regression based object detector. We now complement it with verification tasks of different forms, specifically for corners and within-box foreground. In Section 3.1, We have a brief review of the work flow of RepPoints. A set of representative points are predicted to represent the object through two consecutive regression process, and a conversion function is carried to transform these points into bounding boxes. To increase the compatibility of RepPoints with the auxiliary verification tasks, we first make a small modification to it, such that the first two points of the set $\mathcal{R}$ of predicted points now explicitly represent the top-left and bottom-right corner points. We refer to this as the *explicit-corners* variant. These corner points replace the conversion function used in the original RepPoints, so that the bounding box is defined by these corner points instead of by a *min-max* or *momentum* operation on the point set. With the corner points, the conversion function becomes

$$\mathcal{T}(\mathcal{R}) = \left( \frac{x_1 + x_2}{2}, \frac{y_1 + y_2}{2}, x_2 - x_1, y_2 - y_1 \right). \tag{3}$$

where the four numbers denote $x$-center, $y$-center, width and height, respectively. To this *explicit-corners* variant of RepPoints, we add the auxiliary side-branches for verification. Specifically, we take the feature map right after the $3^{\text{rd}}$ conv layer of the localization head as input, to reuse the existing head for computational savings. As illustrated in Figure 2, a $3 \times 3$ convolutional layer is applied on this feature map, followed by two small sub-networks for the two verification tasks. The corner sub-network consists of a corner pooling layer [16] followed by a $1 \times 1$ conv layer to predict heatmap scores and sub-pixel offsets. The foreground sub-network is a single $1 \times 1$ conv layer to predict the foreground score heatmap. In training, we adopt a multi-task loss:

$$L = L_{\text{RepPoints}} + \lambda_1 L_{\text{corner}} + \lambda_2 L_{\text{foreground}}, \tag{4}$$

with loss weights $\lambda_1 = 0.25$ and $\lambda_2 = 0.1$. More details are given in Appendix.

Customizing the general fusion method of Section 3.3 to RepPoints, we use corner verification for multi-task learning, feature enhancement and joint inference. Foreground verification is instead used only for multi-task learning and feature enhancement. The resulting detector is named RepPoints v2.

### 3.5 Extension to Other Detectors and Problems

The fusion method used for RepPoints can also improve other detectors such as FCOS [29]. As FCOS shares similar classification and localization heads as in RepPoints, the fusion of RepPoints v2 can directly be applied to it. Concretely, the corner and foreground verification heads are applied on the feature map after the $3^{\text{rd}}$ conv layer. The verification output maps are fused into the main branch, and the final regression results are obtained by the joint inference described in Section 3.3.

The fusion method can also be extended to other tasks such as instance segmentation by Dense RepPoints [33], a regression-based method. Since there is an additional object mask annotation, more fine-grained verification formats can be used, such as object contour verification and category-aware semantic segmentation. As shown in Table 7, the additional verification methodology brings 1.3 mask AP gains to Dense RepPoints on the COCO `test-dev` set. More details are presented in Appendix.

## 4 Experiments

We conduct experiments on the challenging MS COCO 2017 benchmark [17], which is split into `train`, `val` and `test-dev` sets with 115K, 5K and 20K images, respectively. We train all the models using the `train` set and conduct an ablation study on the `val` set. A system-level comparison to other methods is reported on the `test-dev` set.

Table 2: Performance of the *explicit-corners* variant of RepPoints.

| variant | +verification | $AP$ | $AP_{50}$ | $AP_{75}$ | $AP_S$ | $AP_M$ | $AP_L$ |
|---|---|---|---|---|---|---|---|
| min-max | | 39.1 | 58.8 | 42.4 | 22.4 | 42.8 | 50.5 |
| | ✓ | 40.7 | 59.8 | 43.7 | 23.3 | 44.4 | 54.0 |
| partial min-max | | 39.0 | 58.7 | 42.4 | 21.8 | 42.5 | 50.7 |
| | ✓ | 40.7 | 59.7 | 43.6 | 23.1 | 44.4 | 54.0 |
| momentum | | 39.1 | 58.9 | 42.2 | 22.3 | 42.6 | 50.8 |
| | ✓ | 40.8 | 59.7 | 43.7 | 23.5 | 44.7 | 53.9 |
| explicit-corners | | 39.1 | 58.8 | 42.5 | 22.4 | 42.6 | 50.6 |
| | ✓ | 41.0 | 59.9 | 43.9 | 23.8 | 44.8 | 54.0 |

Table 3: Ablations on two forms of verification.

| corner | foreground | $AP$ | $AP_{50}$ | $AP_{75}$ | $AP_{90}$ | $AP_S$ | $AP_M$ | $AP_L$ |
|---|---|---|---|---|---|---|---|---|
| | | 39.1 | 58.8 | 42.5 | 14.4 | 22.4 | 42.6 | 50.6 |
| ✓ | | 40.5 | 59.0 | 43.5 | 18.4 | 23.4 | 44.1 | 53.5 |
| ✓ | ✓ | 41.0 | 59.9 | 43.9 | 18.5 | 23.8 | 44.8 | 54.0 |

## 4.1 Implementation Details

We use the mmdetection codebase [2] for experiments. All experiments perform training with an SGD optimizer on 8 GPUs with 2 images per GPU, using an initial learning rate of 0.01, a weight decay of 0.0001 and momentum of 0.9. In ablations, most experiments follow the $1\times$ settings where 12 epochs with single-scale training of $[800, 1333]$ are used, with learning rate decayed by $10\times$ after epoch 8 and 11. Most of the ablations use a ResNet-50 [12] backbone pretrained on ImageNet [25]. We also test our approach using multi-scale ($[480, 960]$) and longer training ($2\times$ settings with 24 epochs in total and the learning rate decayed at epoch 16 and 22) on stronger backbones to see whether the gains by the proposed approaches hold on these stronger baselines.

In inference, unless otherwise specified, we adopt a single-scale test approach with the image size the same as in single-scale training. We also conduct multi-scale testing on the strongest backbone for comparison with the previous state-of-the-art approaches. An IoU threshold of 0.6 is applied for Non-Maximum Suppression (NMS) to remove duplicate boxes.

For RepPoints [32], we use an improved implementation by replacing the IoU assigner with an ATSS assigner [34], yielding 39.1 mAP on COCO `val` using a ResNet-50 model and the $1\times$ settings, 0.9 mAP higher than that reported in the original paper.

## 4.2 Ablation Study

**Explicit-corners variant.** We first validate the effectiveness of the *explicit-corners* variant of RepPoints described in Section 3.4, as shown in Table 2. This variant performs on par with the three variants used in the original RepPoints, but performs 0.2-0.3 mAP better than other variants when the verification module is added. This could be contributed to more effective interaction between verification and regression tasks in this *explicit-corners* variant.

**Forms of verification** Table 3 ablates the two forms of verification. The corner verification task alone brings 1.4 mAP gains over the RepPoints baseline. The benefits are mainly for higher IoU criteria, e.g. $AP_{90}$ is improved by 4.0 mAP while $AP_{50}$ increases by only 0.2 mAP. The additional foreground verification task brings another 0.5 mAP in gains, but mainly on lower IoU criteria, for example, $AP_{50}$ is improved by 0.9 AP while $AP_{90}$ remains about the same.

Table 4: Ablations on three types of fusion.

| multi-task | enhance feature | joint inference | $AP$ | $AP_{50}$ | $AP_{75}$ | $AP_{90}$ | $AP_S$ | $AP_M$ | $AP_L$ |
|---|---|---|---|---|---|---|---|---|---|
| | | | 39.1 | 58.8 | 42.5 | 14.4 | 22.4 | 42.6 | 50.6 |
| ✓ | | | 39.5 | 58.9 | 42.7 | 14.6 | 22.5 | 43.1 | 51.0 |
| ✓ | ✓ | | 40.2 | 60.0 | 43.5 | 15.7 | 24.1 | 43.8 | 52.5 |
| ✓ | ✓ | ✓ | 41.0 | 59.9 | 43.9 | 18.5 | 23.8 | 44.8 | 54.0 |

**Types of fusion** Table 4 ablates the types of fusion, specifically multi-task learning, feature enhancement for regression, and joint inference. Multi-task learning brings a 0.4 mAP gain over the RepPoints baseline. Note that this multi-task learning does not rely on annotations beyond bounding boxes, in contrast to that in Mask R-CNN [10]. The additional feature enhancement operation brings another 0.7 gain. The explicit fusion by joint inference brings increases mAP by 0.8, such that the full approach surpasses its counterpart without verification modules by 1.9 mAP.

**Hyperparamter $r$ in joint inference** The hyperparamter $r$ in joint inference controls the searching range for corner point refinement. $r = 1, 2, 3, 4$ produce mAP of $41.0, 40.8, 40.5, 40.2$, respectively, indicating $r = 1$ performs best. A more sophisticated method for corner point refinement may produce better results but it is not the main focus of this work.

**Complexity analysis and runtime.** Our approach involves slightly more parameters (38.3M vs 37.0M) and marginally more computation (244.2G vs 211.0G) than the original RepPoints. This overhead mainly occurs at the additional heads to produce verification score/offset maps. We also conduct RepPoints with heavier computation, by adding one more convolutional layers on the heads, resulting in a baseline with similar parameters and computations as our approach (38M/235.8G v.s. 38.3M/244.2G). The enhanced baseline model performs 0.2 mAP better than the vanilla RepPoints, indicating that the improvements by our approach are mostly not due to more parameters and computation. For real inference speed, the speed of RepPoints v1 is 12.7 FPS (img/s) using ResNet-50 on a Titan XP GPU, while that of RepPoints v2 is 10.1 FPS. With a ResNeXt-101-DCN backbone, the speeds are 4.3 FPS v.s. 3.8 FPS for RepPoints v1 and v2, respectively.

Table 5: Experiments on RepPoints baselines with stronger backbones using $2\times$ settings (24 epochs) and multi-scale training ([480, 960]) on COCO val set.

| backbone | +verification | $AP$ | $AP_{50}$ | $AP_{60}$ | $AP_{70}$ | $AP_{80}$ | $AP_{90}$ |
|---|---|---|---|---|---|---|---|
| ResNet-50 | | 41.8 | 61.8 | 58.1 | 51.1 | 38.6 | 15.9 |
| | ✓ | 43.9 | 63.1 | 59.3 | 52.5 | 40.1 | 20.6 |
| ResNet-101 | | 43.4 | 63.3 | 59.4 | 53.0 | 40.4 | 18.0 |
| | ✓ | 45.5 | 64.5 | 60.6 | 54.1 | 42.3 | 22.2 |
| ResNeXt-101 | | 45.5 | 65.9 | 62.1 | 55.2 | 42.4 | 19.7 |
| | ✓ | 47.3 | 66.9 | 62.9 | 56.1 | 44.0 | 23.7 |

**Stronger baselines.** We further validate our method on stronger RepPoints baselines, using longer/multi-scale training ($2\times$ settings) and stronger backbones, as shown in Table 5. It can be seen that the gains are well maintained on these stronger RepPoints baselines, at about 2.0 mAP. This indicates that the proposed approach is largely complementary to improved baseline architecture, in contrast to many techniques that have exhibited decreasing gains with respect to stronger baselines.

Table 6: Applying the verification module to FCOS, which is implemented in mmdetection.

| | backbone | $AP$ | $AP_{50}$ | $AP_{75}$ | $AP_{90}$ | $AP_S$ | $AP_M$ | $AP_L$ |
|---|---|---|---|---|---|---|---|---|
| FCOS | ResNet-50 | 38.2 | 57.1 | 41.2 | 15.3 | 22.2 | 42.3 | 49.5 |
| +verification | ResNet-50 | 39.5 | 57.7 | 41.9 | 18.4 | 22.3 | 43.2 | 52.7 |

Table 7: Adding the verification module to the instance segmentation algorithm Dense RepPoints on COCO `test-dev`.

| | backbone | $AP_{\text{mask}}$ | $AP_{50}$ | $AP_{75}$ | $AP_S$ | $AP_M$ | $AP_L$ |
|---|---|---|---|---|---|---|---|
| Dense RepPoints | ResNet-50 | 37.6 | 60.4 | 40.2 | 20.9 | 40.5 | 48.6 |
| +verification | ResNet-50 | 38.9 | 61.5 | 41.9 | 21.2 | 42.0 | 51.1 |

**Visualization.** The visualization results are given in Appendix.

### 4.3 Comparison to State-of-the-art Methods

We compare the proposed method to other state-of-the-art object detectors on the COCO2017 `test-dev` set, as shown in Table 8. We use GIoU [24] loss instead of smooth-l1 loss in the regression branch here. With ResNet-101 as the backbone, our method achieves 46.0 mAP without bells and

Table 8: Comparison of RepPoints v2 to state-of-the-art detectors on COCO `test-dev`. * denotes that the number is obtained by multi-scale testing.

| method | backbone | epoch | $AP$ | $AP_{50}$ | $AP_{75}$ | $AP_S$ | $AP_M$ | $AP_L$ |
|---|---|---|---|---|---|---|---|---|
| RetinaNet [19] | ResNet-101 | 18 | 39.1 | 59.1 | 42.3 | 21.8 | 42.7 | 50.2 |
| FCOS [29] | ResNeXt-101 | 24 | 43.2 | 62.8 | 46.6 | 26.5 | 46.2 | 53.3 |
| DCN V2* [38] | ResNet-101+DCN | 18 | 46.0 | 67.9 | 50.8 | 27.8 | 49.1 | 59.5 |
| RepPoints* [32] | ResNet-101+DCN | 24 | 46.5 | 67.4 | 50.9 | 30.3 | 49.7 | 57.1 |
| MAL* [14] | ResNeXt-101 | 24 | 47.0 | 66.1 | 51.2 | 30.2 | 50.1 | 58.9 |
| FreeAnchor* [35] | ResNeXt-101 | 24 | 47.3 | 66.3 | 51.5 | 30.6 | 50.4 | 59.0 |
| ATSS* [34] | ResNeXt-101+DCN | 24 | 50.7 | 68.9 | 56.3 | 33.2 | 52.9 | 62.4 |
| TSD* [27] | SENet154+DCN | 24 | 51.2 | 71.9 | 56.0 | 33.8 | 54.8 | 64.2 |
| CornerNet [16] | HG-104 | 100 | 40.5 | 56.5 | 43.1 | 19.4 | 42.7 | 53.9 |
| ExtremeNet [37] | HG-104 | 100 | 40.2 | 55.5 | 43.2 | 20.4 | 43.2 | 53.1 |
| CenterNet [6] | HG-104 | 100 | 44.9 | 62.4 | 48.1 | 25.6 | 47.4 | 57.4 |
| RepPoints v2 | ResNet-50 | 24 | 44.4 | 63.5 | 47.7 | 26.6 | 47 | 54.6 |
| RepPoints v2 | ResNet-101 | 24 | 46.0 | 65.3 | 49.5 | 27.4 | 48.9 | 57.3 |
| RepPoints v2 | ResNeXt-101 | 24 | 47.8 | 67.3 | 51.7 | 29.3 | 50.7 | 59.5 |
| RepPoints v2 | ResNet-101+DCN | 24 | 48.1 | 67.5 | 51.8 | 28.7 | 50.9 | 60.8 |
| RepPoints v2 | ResNeXt-101+DCN | 24 | 49.4 | 68.9 | 53.4 | 30.3 | 52.1 | 62.3 |
| RepPoints v2* | ResNeXt-101+DCN | 24 | **52.1** | 70.1 | 57.5 | 34.5 | 54.6 | 63.6 |

whistles. By using stronger ResNeXt-101 [31] and DCN [3] models, the accuracy rises to 49.4 mAP. With additional multi-scale tests as in [34], the proposed method achieves 52.1 mAP.

### 4.4 Extension to Other Detectors and Applications

**Direct application to FCOS** FCOS [29] is another popular regression based object detector. We directly apply our approach without modification to this detector, and 1.3 mAP improvements are obtained as shown in Table 6, indicating generality of the proposed approach.

**Extension to instance segmentation** Table 7 shows the effect of additional verification modules in the instance segmentation method of Dense RepPoints [33]. The additional contour and foreground modules improve accuracy by 1.3 mAP, demonstrating the broad applicability of the fusion method.

## 5 Conclusion

In this paper, we propose *RepPoints v2*, which enhances the original regression-based *RepPoints* by fusing verification tasks in various ways. A new variant of RepPoints is proposed to increase the compatibility with the auxiliary verification tasks. The resulting object detector shows consistent improvements over the original RepPoints under different backbones and training approaches. It also achieves 52.1 mAP on the COCO `test-dev`. Moreover, this approach could be easily transferred to other detectors and the instance segmentation domain, boosting the performance of the base detector/segmenter by a considerable margin.

## Acknowledgments and Disclosure of Funding

This work was supported by National Key R&D Program of China (2018YFB1402600), Key-Area Research and Development Program of Guangdong Province (No. 2019B121204008)] and Beijing Academy of Artificial Intelligence.

## Broader Impact

Since this work is about designing better object detectors, researchers and engineers engaged in object detection and instance segmentation on natural images, medical images and even video data may benefit from this paper. If there is any failure in this system, the model may not detect objects correctly. Similar to most object detectors, the detection results may not be interpretable, thus it is hard to predict failure scenarios. This object detector also leverages biases in the dataset used for training, and may incur a performance drop on datasets which have a domain gap with the training dataset.

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
