[Supplementary Material · supplemental-4400.pdf]

# Supplementary Materials for RepPoints V2

## A  Evaluation Code

The directory "code_for_RepPointsV2" contains code for evaluating our method. Please follow README.md to run the code.

## B  Details of Verification Tasks

### B.1  Corner Point Verification

**Ground-truth assignment.**  We follow CornerNet [1] to assign ground-truth corners. For each corner, only the corner itself is positive location, and all other locations are negative. Moreover, the penalty given to negative locations within a radius of the positive location is reduced. Specifically, for a given corner point $p = (p_x, \ p_y)$ on original image, the size of the ground-truth heatmap $Y$ with $s\times$ downsampled rate is $\frac{H}{s} \times \frac{W}{s}$, and the corresponding location of $p$ on $Y$ is $\hat{p} = \lfloor \frac{p}{s} \rfloor$. The penalty weight of negative locations is defined as a inverse Gaussian function:

$$Y_{xy} = \exp\left(-\frac{(x - \hat{p}_x)^2 + (y - \hat{p}_y)^2}{2\sigma_p^2}\right) \tag{1}$$

where $\sigma_p$ is an object size-adaptive standard deviation, $x$ and $y$ indicate the location of a negative point. Note that for different positive points, the penalty weight of a negative point may be different. Therefore, the largest one as the penalty weight of the negative point.

For additional offset prediction, we follow [1] that only supervises the positive locations. For a given corner point $p$ and its corresponding downsampled location $\hat{p}$, the training target is:

$$o(\hat{p}) = \left(\frac{p_x}{s} - \left\lfloor \frac{p_x}{s} \right\rfloor, \ \frac{p_y}{s} - \left\lfloor \frac{p_y}{s} \right\rfloor\right) \tag{2}$$

**Loss.**  Follow CornerNet [1], we use a modified focal loss [2] to learn the corner heatmap. The loss is defined as

$$L_{\text{heatmap}} = \frac{-1}{N} \sum_{i=1}^{H} \sum_{j=1}^{W} \begin{cases} (1 - p_{ij})^\alpha \log(p_{ij}) & \text{if } y_{ij} = 1 \\ (1 - y_{ij})^\beta (p_{ij})^\alpha \log(1 - p_{ij}) & \text{otherwise} \end{cases} \tag{3}$$

where $N$ is the number of objects in an image, $p_{ij}$ and $y_{ij}$ are the score and label at location $(i, j)$ in the predicted heatmap. We set $\alpha = 2$ and $\beta = 4$, following [1].

In addition, the loss to learn offset are defined as:

$$L_{\text{offset}} = \frac{1}{N} \sum_{k=1}^{N} \text{SmoothL1Loss}\left(o(\hat{p}_k), \hat{o}(\hat{p}_k)\right) \tag{4}$$

where $o$ is the groundtruth offset, $\hat{o}$ is the predicted offset, $\hat{p}_k$ is the $k$-th corner point. Finally, the overall loss $L_{\text{corner}}$ of corner branch is simply defined as the summation of $L_{\text{heatmap}}$ and $L_{\text{offset}}$.

### B.2  Within-box Foreground Verification

**Normalized focal loss.**  The normalized focal loss is defined as:

Table 1: Adding the verification module to the instance segmentation algorithm Dense RepPoints on COCO `test-dev`.

| | backbone | $AP_{\text{mask}}$ | $AP_{50}$ | $AP_{75}$ | $AP_S$ | $AP_M$ | $AP_L$ |
|---|---|---|---|---|---|---|---|
| Dense RepPoints | ResNet-50 | 37.6 | 60.4 | 40.2 | 20.9 | 40.5 | 48.6 |
| +contour&fg | ResNet-50 | 38.6 | 61.4 | 41.7 | 21.3 | 41.8 | 50.8 |
| +joint inference | ResNet-50 | 38.9 | 61.5 | 41.9 | 21.2 | 42.0 | 51.1 |

$$\mathcal{L}_{\text{fg}} = \sum_{c=1}^{C}\sum_{i=1}^{H}\sum_{j=1}^{W}\begin{cases} \frac{-1}{N_W}w_{cij}\cdot\alpha\left(1-p_{cij}\right)^{\gamma}\log\left(p_{cij}\right) & \text{if } y_{cij}=1 \\[2mm] \frac{-1}{N}\left(1-\alpha\right)\left(p_{cij}\right)^{\gamma}\log\left(1-p_{cij}\right) & \text{otherwise} \end{cases} \tag{5}$$

where $y_{cij}$ is the value on the ground-truth foreground heatmap, $p_{cij}$ is the $c$-th category score at location $(i,j)$ of the predicted heatmap, $w_{cij}$ is the normalizing factor, which is defined as:

$$w_{cij} = \begin{cases} \frac{1}{S_{cij}} & \text{if } y_{cij}=1 \\[2mm] 0 & \text{otherwise} \end{cases} \tag{6}$$

where $S_{cij}$ is the area of the object that $(i,j)$ lies in. If multiple objects of the same category collide at the same location, we would take the smallest size. $N_W$ is defined as $\sum_{c=1}^{C}\sum_{i=1}^{H}\sum_{j=1}^{W} w_{cij}$, the sum of normalizing factor at all locations. $N$ is the number of positive points. $\alpha$ and $\gamma$ is set as 0.25, 2, respectively.

### B.3 Overall Loss

The overall loss is defined as:

$$L = L_{\text{RepPoints}} + \lambda_1 L_{\text{corner}} + \lambda_2 L_{\text{fg}}, \tag{7}$$

and $\lambda_1 = 0.25$ and $\lambda_2 = 1.0$. $L_{\text{corner}}$ and $L_{\text{fg}}$ are defined above. We briefly review $L_{\text{RepPoints}}$. It is defined as

$$L_{\text{RepPoints}} = L_{\text{cls}} + \gamma_1 L_{\text{box}_1} + \gamma_2 L_{\text{box}_2} \tag{8}$$

where $L_{\text{cls}}$ is classification loss defined as focal loss, $L_{\text{box}_1}$ is the localization loss of the initial stage, $L_{\text{box}_2}$ is for the refine stage. Both are SmoothL1 loss. $\gamma_1$ and $\gamma_2$ are set as 0.5 and 1, respectively.

## C  Extension to Instance Segmentation

**Training settings.** We based on Dense RepPoints [5] to validate the effectiveness of our method, due to the Dense RepPoints is the state-of-the-arts regression-based instance segmentation approach. Because the contour points has no type, only one heatmap is used for predicting all contour points. Other parameters, network architectures and training details are same as object detection.

**Joint inference.** With only a few modifications, joint inference can also be used for instance segmentation. For a predicted representative point, if it is close to the contour point, then we refine the predicted representative point set by adding the adjacent contour point into the set. More specifically, if the score of representative point in the contour heatmap is greater than 0.5, then the point with the highest contour score among all the points with a distance less than 1 are added to the set.

**Experimental results.** The results is given in Table 1. ResNet-50 backbone and 3x scheduler are adopted. By adding verification module, the performance are elevated by 1.0 mAP, further applying the joint inference, additional 0.3 mAP is improved. This demonstrates the flexibility of our proposed method.

## D Visualization

Figure 1 shows some object detection results comparison on COCO 2017 [3] between *RepPoints v1* [4] and *RepPoints v2*. Both methods adopt ResNet-50 backbone and 1x scheduler. As can be seen, compared to *RepPoints v1*, *RepPoints v2* could provide us more precise localization results.

Figure 2 gives the visualization of main component of *RepPoints v2*. From left to right are set of representative points predicted, foreground prediction, top-left corner prediction and bottom-right corner prediction. As can be seen, all components could provide informative cues, leading to better performance.

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

Figure 1: Visualization results of *RepPoints v1* and *RepPoints v2*. image on the top row is the detection of *RepPoints v1* and the bottom row is for *RepPoints v2*. The red boxes are generated without joint inference while green boxes adopts joint inference. As can be seen, our full version of *RepPoints v2* could achieve better localization results.

Figure 2: Visualization of main component of *RepPoints v2*. From left to right are set of representative points predicted, foreground prediction, top-left corner prediction and bottom-right corner prediction.