[Reviews · NeurIPS 2020]

Review 1

Summary and Contributions: This paper introduces classification task into the RepPoints framework. The specific contributions include proposing to model verification tasks by auxiliary side-branches, and combining them at the feature level and inference phase. The final model obtains 52.1 mAP on COCO test-dev by a single model. And It also shows effectiveness on instance segmentation.

Strengths: The paper provides detailed description of their proposed methods, including the implementation details and the evaluation codes. Extended experiments also prove the effectiveness of their methods. This paper is aimed at object detection, which is a basic task in the field of computer vision, and has a high relevance to the NeurIPS community.

Weaknesses: The novelty of this paper is limited. The idea of incorporating other tasks into the framework of object detection framework to enjoy the improvements brought by multi-task learning is not novel. Although authors state that their method does not require additional annotations, while mask r-cnn does. But the subtle differences between these are not very innovative. There are also several works using keypoints to assist object detection, such as CentripetalNet [1] and RetinaFace [2]. Reference: [1] Dong Z, Li G, Liao Y, et al. Centripetalnet: Pursuing high-quality keypoint pairs for object detection. CVPR 2020. [2] Deng J, Guo J, Zhou Y, et al. Retinaface: Single-stage dense face localisation in the wild.

Correctness: The method proposed in the article sounds effective, and the experimental part fully proves it.

Clarity: The paper is well written and easy to read.

Relation to Prior Work: The paper discuss clearly the difference between RepPoints v1 and v2. But the difference with other similar works need to be claimed detailedly.

Reproducibility: Yes

Additional Feedback: The rebuttal partly address my concerns, I would like to raise my score to 6.


Review 2

Summary and Contributions: This paper propose the idea of introducing verification tasks into regression based object detectors. The authors discuss two kinds of verification tasks, corner point verification and within-box foreground verification and introduce several fusion methods to combine the advantages of verification and regression.

Strengths: 1. This paper is well-written and easy to follow for readers who are familiar with RepPoints. 2. The finding of differences between verification based method and regression based method is very interesting. 3. This paper introduce a joint inference method to combine the advantages of verification and regression, which yield significant performance improvement.

Weaknesses: 1. It will be better to add more descriptions about RepPoints. It is hard for readers to follow the implementation details described in Sec 3.4 if they are not familiar with RepPoints. What is the meaning of “..., such that the first two points explicitly represent the top-left and bottom-right corner points.” ? In my understanding, the authors still predict n sample points following RepPoints, but they adopt the first two points to do point-bbox transformation. I suggest the authors replace the description of “the first two points” with “the first two points in point sets R and R`”. Describe the definition of RepPoints loss in the paper instead of appendix will make it more easy to understand the implementations. 2. Within-box foreground verification plays a similar role as centerness proposed by FCOS. The authors should add discussion and comparison. 3. The major improvement in higher IOU criteria is obtained by the corner point head. It would be better if the author could explain why corner point based method can yield more accurate localization performance. 4. Corner point verification and joint inference bring significant performance improvement for higher IOU criteria. However, as shown in Table 1. CornerNet still outperform RepPoints v2 4.9 mAP in AP90. I guess this is because CornetNet adopt high resolution feature maps to obtain more accurate results. Would the authors explain the reason why adopting high resolution feature map for corner point verification could not yield better performance?

Correctness: Yes

Clarity: Yes

Relation to Prior Work: Yes

Reproducibility: Yes

Additional Feedback: See above ========================= The rebuttal addresses most of my concerns, I will keep as weak accept.


Review 3

Summary and Contributions: The paper proposes to use verification branches (i.e., heatmap regression/classification) to improve the performance of regression-based object detectors such as RepPoints and FCOS. The knowledge learned by the verification branches is fused into the detectors at both feature-level and result-level. The proposed method consistently improves RepPoint by about 2 mAP and FCOS by more than 1 AP on COCO.

Strengths: 1) The proposed method of using verficaition to improve the regression-based detectors sounds reasonable. I agree that the verfication and regression tasks are complementary, as shown in the paper. 2) The proposed method consistently improves the preformance of previous regression-based detectors, which shows the its effectiveness. 3) It also achieves state-of-the-art performance on MS-COCO.

Weaknesses: 1) It is more common and accurate to use the term "heatmap classification/regression" rather than "verification". 2) It is not suitable to call the work RepPoints v2, as the main concern is to add extra heatmap classification tasks to improve the precision of regression. Moreover, if I understand the paper correctly, in L201, the paper only makes use of two corners to represent an object (i.e., the explicit corners variant). Using explict corners is in conflict with the motivation of the original RepPoints paper, which attempts to use multiple points to represent an object. 2) In addition, the hyperparameter r controls the search range for refining. Please provide detailed ablations on the hyperparameter.

Correctness: Yes.

Clarity: The writing is good but it can be improved.

Relation to Prior Work: Yes.

Reproducibility: Yes

Additional Feedback: The authors address my questions in rebuttal, so I keep my score.


Review 4

Summary and Contributions: This paper analyzes the verification and regression methodologies in current object detection systems and proposes RepPoint v2. The authors incorporate anchor-based and point-based verification tasks as auxiliary tasks to facilitate the training of RepPoint. Jointly inferencing with the auxiliary tasks also boost the performance further more. The model obtains about 2.0 mAP improvements on COCO test-dev benchmark. The authors also prove the generalization on other detectors and instance segmentation applications.

Strengths: 1. This method is well-written and easy to follow. 2. The experiments are thorough and verify the effectivenss of the proposed method. 3. Overall, I think this work is a great complementary to RepPoint. RepPoint v2 remedy the drawbacks of direct regression in RepPoint and boost the performance. As mentioned in the literature, all most other objects do several steps to regression the localization targets, while reppoint do one-shot regression which is inferior to other methods. This paper propose methods to remedy the drawbacks. The motivation is reasonable and the problem is tackled with good methods.

Weaknesses: 1. The descriptions in joint inference is not very clear. I cannot get how the refine process do according to Equ. 2. It would be great if the authors can clear this part during the rebuttal and polish this part in the final version. 2. I have some doubts about the definations in Table1. What's the different between anchor-based regression and the regression in RepPoints? in RetinaNet, there is also only a one-shot regression. And in ATSS, this literature has proved that the regression methods do not influence a lot. The method that directly regresses [w, h] to the center point is good enough. While RepPoints regresses distance to the location of feature maps. I think there is no obvious difference between the two methods. I hope the authors can clarify this problem. If not, the motivations here is not solid enough. 3. It would be great if the authors can analyze the computational costs and inference speeds for the proposed method.

Correctness: Yes.

Clarity: Yes, the paper is well written.

Relation to Prior Work: Yes.

Reproducibility: Yes

Additional Feedback: Comments after author feedback: As the paper receives positive comments and the rebuttal address the comments. I keep my original review to accept the paper. But I hope the authors can polish the academic writing of the paper especially the technical details.

[Author Response · NeurIPS 2020]

We thank the reviewers for their insightful and constructive comments.

**[R1] Multi-task not novel.** We would like to clarify that our contribution is far beyond multi-task learning: a unified
and general framework with 3 mechanisms to take advantage of verification tasks to help regression methods. In fact,
the multi-task learning only contributes +0.4 mAP gains, while others contribute +1.5 mAP gains (see Tab. 4).

**[R1] several works using keypoints, e.g. CentripetalNet and RetinaFace.** First, our method is beyond using
corner/keypoint verification: it generally exploits various verification tasks to help regression methods, e.g. the
foreground verification task, which is not exploited by CentripetalNet and RetinaFace. Second, even considering
corner verification alone, our work is significantly different from CentripetalNet and RetinaFace: CentripetalNet is an
improvement of CornerNet with a better corner matching mechanism, which is purely verification-based. In contrast,
our method focuses on how to combine corner verification into regression methods to improve object localization.
RetinaFace requires explicit keypoint annotations and only exploits multi-task learning, while our method does not
require additional annotation and is far beyond pure multi-task learning. We will discuss these works in revision.

**[R2] Add more description about RepPoints; still predict n sample points?** Thanks for the suggestion and we will
rewrite it in the revision. Yes, the understanding is correct.

**[R2] Within-box verification and FCOS's centerness.** We conduct experiments to include the verification task of
FCOS's centerness. On a 40.5 AP baseline (+corner in Tab. 3 of this paper), an additional centerness branch achieves
40.6 mAP which has almost no gains over the baseline, while our additional foreground branch has +0.5 mAP gains
(41.0 mAP vs. 40.5 mAP). On a 41.0 mAP baseline (+corner+foreground in Tab. 3 of this paper), an additional
centerness branch achieves 41.0 mAP, which has no gains over the baseline. Actually, although FCOS's centerness and
our within-box verification share some similarity, they are designed for different roles: while FCOS's centerness aims
for adjusting object scores and weighting the regression loss, the within-box verification aims to complement the main
regression branch. We will add discussions in the revision.

**[R2] Explain why corner verification produces more accurate localization results.** As described in Line 32-36,
there may be two probable intuitive explanations: corner points represent the exact spatial extent of bounding box; each
feature in corner point verification is well aligned to the corresponding point.

**[R2] Why CornerNet outperforms RepPoints v2 in AP90.** It is difficult to rigorously analyze all the factors for the
performance gap in AP90 due to too many implementation differences. Nevertheless, we think there are probably two
main factors: 1) our backbone is different (Res50 v.s HG-104), and some works have shown that ResNet architectures
perform significantly worse than Hourglass ones in detecting corner points, e.g. 30.2 (R101) vs. 38.4 (HG-104) in
Tab. 4 of CornerNet; 2) our highest resolution (C3) is lower than that of CornerNet (C2). A possible direction towards
better AP90 is to use HG architectures and higher resolution, but it would significantly change the main branch of
RepPoints (built on ResNet-FPN-C3-C7 structure), and we will leave it as our future works.

**[R2] Why higher resolution does not yield better performance.** The corner heatmap is used in both feature-level
fusion and result-level joint inference. For feature-level fusion, we compared C3 heatmap with C3-C7 heatmaps, where
we find the latter performs better. We hypothesize it is because more positive samples by C3-C7 benefit the training. For
result-level joint inference, higher resolution does yield better performance that all levels using C3 heatmap performs
better than that using the corresponding heatmap level. We will add the discussion in revision.

**[R3] Terminologies.** Thanks for the suggestion. We will carefully examine and revise them.

**[R3] Ablation about hyperparameter $r$.** $r = 1, 2, 3, 4$ produce mAP of $41.0, 40.8, 40.5, 40.2$, respectively, indicating
$r = 1$ performs best. We will add this ablation in revision.

**[R4] Details of joint inference.** Intuitively, for a regressed corner point $p^t$, Eq. (2) means searching a point with the
highest corner heatmap score in a neighborhood of $r$ as its final location. Regressed points on all pyramidal levels use
the C3 corner heatmap for searching, and $r=1$ corresponds to a 8-pixel neighborhood. We will polish the description.

**[R4] Doubts about definitions in Tab.1.** We agree with the reviewer's comment that the relatively smaller displacement
regression targets in RetinaNet does not have benefit over RepPoints (or FCOS) which regress the box extent from the
center point. However, this has no contradiction with our categorization of RetinaNet as an verification+regression
framework and RepPoints as a regression framework: there are pre-defined anchors (coarse localization hypothesis) in
RetinaNet, while there is no such coarse hypothesis for RepPoints (anchor-free). Moreover, our Paragraph 2 delivers
the same information as the reviewer's comment, which motivates us to clarify the difference between two verification
methods (CornerNet vs. RetinaNet, see Paragraph 3) and to further propose our method (Paragraph 4).

**[R4] Computational costs and inference speeds.** FLOPs comparison is described in Line 265-271. For real inference
speed, the speed of RepPoints v1 is 12.7 FPS (img/s) using ResNet-50 on a Titan XP GPU, while that of RepPoints
v2 is 10.1 FPS. With a ResNeXt-101-DCN backbone, the speeds are 4.3 FPS v.s. 3.8 FPS for RepPoints v1 and v2,
respectively. We will add inference speed comparisons in the revision.

[Meta-Review · NeurIPS 2020]

The reviewers overall found merit in the main idea of this paper (introducing verification tasks in the setting of regression-based object detection). Overall, there is a good amount of analysis on differences between verification based methods as well as the introduction of a new method that does well. The reviewers found this work to be a good complement to RepPoints. I encourage the authors to take into account the feedback regarding the writing and the lack of some details for the final version of this work.